# Application of Near-Infrared Optical Feedback Cavity Enhanced Absorption Spectroscopy (OF-CEAS) to the Detection of Ammonia in Exhaled Human Breath

**DOI:** 10.3390/s19173686

**Published:** 2019-08-25

**Authors:** Zhifu Luo, Zhongqi Tan, Xingwu Long

**Affiliations:** Department of Optoelectronic Engineering, College of Advanced Interdisciplinary Studies, National University of Defense Technology, Changsha 410073, China

**Keywords:** optical feedback, ammonia, cavity enhanced absorption spectroscopy, trace gas detection, breath gas analysis

## Abstract

The qualitative and quantitative analysis to trace gas in exhaled human breath has become a promising technique in biomedical applications such as disease diagnosis and health status monitoring. This paper describes an application of a high spectral resolution optical feedback cavity enhanced absorption spectroscopy (OF-CEAS) for ammonia detection in exhaled human breath, and the main interference of gases such as CO_2_ and H_2_O are approximately eliminated at the same time. With appropriate optical feedback, a fibered distributed feedback (DFB) diode laser emitting at 1531.6 nm is locked to the resonance of a V-shaped cavity with a free spectral range (FSR) of 300 MHz and a finesse of 14,610. A minimum detectable absorption coefficient of *α*_min_ = 2.3 × 10^−9^ cm^−1^ is achieved in a single scan within 5 s, yielding a detection limit of 17 ppb for NH_3_ in breath gas at low pressure, and this stable system allows the detection limit down to 4.5 ppb when the spectra to be averaged over 16 laser scans. Different from typical CEAS with a static cavity, which is limited by the FSR in frequency space, the attainable spectral resolution of our experimental setup can be up to 0.002 cm^−1^ owing to the simultaneous laser frequency tuning and cavity dither. Hence, the absorption line profile is more accurate, which is most suitable for low-pressure trace gas detection. This work has great potential for accurate selectivity and high sensitivity applications in human breath analysis and atmosphere sciences.

## 1. Introduction

Exhaled breath gas analysis has a long history and is useful for medical diagnostic and health state monitoring. It has been well known since thousands of years that the exhaled breath of people suffering from certain diseases has a particular smell. In recent years, breath gas analysis combined with some new approaches have been established for scientific studies owing to its noninvasive features, simple sample preparation and harmless nature. More than 1000 different molecules have been demonstrated in human breath gas by a large number of studies [1,2]. For modern breath analysis, a milestone was achieved by Linus Pauling in 1971 when he identified large numbers of volatile organic compounds (VOCs) in exhaled human breath, most of them at very low concentrations (at the level of part per billion (ppb) or even part per trillion (ppt) ) [3]. Some specific gases are confirmed as the biomarkers of corresponding diseases: for example, alkanes appear at an elevated concentration levels in the exhaled breath of patients who have lung cancer [4]; formaldehyde is considered as a biomarker of breast cancer [5]; and excess acetone appears in the exhaled breath of patients with Type 1 diabetes [6]. Breath gas measurement is potentially useful for biomedical diagnosis and health condition monitoring, however, owing to the large number of compounds and their ultra-low concentrations in exhaled human breath, a detector with high sensitivity and accurate selectivity is required for the qualitative and quantitative analysis of the biomarkers. Up to now, a variety of traditional gas analysis techniques have been applied to the measurement of human breath, such as chemiluminescence, gas chromatography (GC), and mass spectrometry (MS) [7]. These methods can achieve detection limits of ppb or even ppt, and realize the simultaneous analysis of multiple compounds, however, GC and MS require expensive instruments and complicated sample preparation, so real-time measurements cannot be realized. Recently, methods based on laser absorption spectroscopy show a vibrant development in trace gas detection applications. For example, Karpf et al. have presented a simplified cavity ringdown setup for trace gas detection [8]. A high power, broad wavelength range, multi-mode Fabry–Perot diode laser is applied as light source, and a detection limit of 530 ppt for nitrogen dioxide (NO_2_) in zero air is achieved in a single measurement. Tomberg demonstrated the use of cantilever-enhanced photoacoustic spectroscopy in trace gas detection, where sub-ppt level sensitivity is obtained during the measurement of a strong absorption cross-section of hydrogen fluoride [9]. Furthermore, simultaneous measurement of C_2_H_6_ and CH_4_, and δ^13^C-CH_4_ isotope ratio was presented by Loic et al. using a mid-infrared CEAS technique, and a minimum absorption coefficient of 2.8 × 10^−9^ cm^−1^ is acquired in a single scan [10].

Ammonia (NH_3_) is a familiar gas in exhaled human breath, and its concentration for a healthy human is regularly in the 0–2.0 ppm range. Excess concentration of ammonia is observed in the breath gas of patients suffering from *Helicobacter pylori* infections [11], renal failure, oral cavity disease [12] and mental disorders [13]. Early in 1987, the laser absorption spectroscopy technique applying a tunable diode laser was used to detect the concentration of NH_3_ in exhaled breath by Lachish et al. [14]. They matched a mid-infrared lead-salt light source to an absorption line of NH_3_ at 10.74 µm (930.76 cm^−1^). The length of the sample cell is 50 cm, a detection limit of 1 ppm was achieved in an integration time of about 10 s and thus near real-time measurement was achieved. With the development of mid-infrared radiation laser sources, such as the quantum cascade laser (QCL), Manne et al. [15] reported a measurement of NH_3_ in exhaled human breath based on the cavity ring-down spectroscopy (CRDS) technique. A thermo-electrically cooled DFB-QCL emitting at 10.3 µm (970.8 cm^−1^) was applied as the light source, and the system achieved a minimum detection sensitivity of 50 ppb for NH_3_ in a single scan within 20 s. The CEAS using the optical frequency comb (OFC) as a light source was established by the Ye group to achieve the measurements of multiple components simultaneously [16]. In their study, a spectral region centered at 1.51 µm (6,623 cm^−1^) was applied, and they achieved a detection limit of 18 ppb for NH_3_ in human breath, where a standard sample with 4.4 ppm NH_3_ in N_2_ was used as a reference to eliminate the interferences of water vapor and carbon dioxide.

As a result of quantum transitions, absorption spectra of molecules are generated between molecular energy levels. In the mid-infrared region, the absorption band comes from the change of the molecule’s rotational and/or vibrational energy state [17]. These spectra are mainly composed of separated narrow lines, and a single line has the Voigt profile, which is a convolution of Gaussian and Lorentzian profiles. Based on the theories of spectral distribution and the line shape, it is possible to selectively detect a gas molecule in a low concentration and even in the presence of interfering gases. It should be noted that overtone transitions lie in the near-infrared band are one to two orders of magnitude less intense. However, in the near-infrared spectral region, the DFB semiconductor lasers and photodetectors are commercially available owing to the development needs of telecommunications, and these devices also perform higher capability at room temperature compared with the devices working in the middle-infrared region. In order to obtain available trace gas detection, it is imperative to match the suitable sensor parameters to the absorption spectrum of the given gas.

The fundamental theory dominating absorption spectroscopy is the Beer–Lambert law. The attenuation of optical radiation at wavelength *λ* when passing through an absorption sample is generally described by the absorption coefficient *α*, and the change of radiation intensity is shown as:(1)$$I(\lambda )=I_{0}(\lambda )\exp[-\alpha (\lambda )L],$$
where *I*_0_(*λ*) and *I*(*λ*) are the intensities of incident radiation and emergent radiation, respectively, and *L* is the optical path in the absorption sample. In this formula, *α* can be described as:(2)$$\alpha (\lambda )=N\sigma (\lambda )=NS_{i}g(\lambda -\lambda_{0}),$$
here, *N* is the concentration of absorption gas molecules, *σ*(*λ*) and *S_i_* represent the absorption cross-section and the line strength of the gas molecule, and *g*(*λ* − *λ*_0_) is the normalized line profile.

CEAS is now regarded as one of the most promising techniques for real-time trace gas detection, with a high finesse cavity as the sample cell, CEAS can reach a 100–10000 times effective absorption path, yielding an enhancement of the detection sensitivity. In this paper, we report on the measurement of NH_3_ in exhaled human breath, which is accomplished by a fibered optical feedback cavity enhanced near-infrared absorption spectrometer. In the very beginning, this technique was introduced by Morville et al. in 2005 [18], with robust and compact instruments, it allows qualitative and quantitative analysis to trace gas and is suitable for in-situ applications. Taking advantage of optical feedback (OF) permits OF-CEAS to realize the efficient injecting of laser light into a high finesse resonant optical cavity [19], which is extremely difficult for conventional CEAS and CRDS, therefore, this technique can effectively improve the spectral resolution and provide a better signal-to-noise ratio (S/N) [20]. We will describe our OF-CEAS experimental setup exploited with a 50 cm long V-shaped cavity (FSR = 300 MHz) in detail, which is different from the normal CEAS and CRDS sensors applying static cavity. A detection limit of 17 ppb for NH_3_ in breath gas is achieved at low sample pressure in a single scan within 5 s, and the attainable spectral resolution of this instrument can be better than 0.002 cm^−1^.

## 2. Materials and Methods

### 2.1. Optical Feedback Cavity Enhanced Absorption Spectroscopy (OF-CEAS)

The detection limit of ordinary spectrophotometry is about 10^−4^ cm^−1^ [21], while the CEAS techniques can achieve a detection limit up to 10^−9^ cm^−1^ [22]. CEAS has similar operation mechanism as CRDS. In both techniques, the essential component of the experimental system is a high finesse resonant optical cavity built with high reflectivity mirrors, and the high-quality cavity generates a long effective optical length, which can be thousands of times longer than the cavity length [23]. As an improvement of CEAS, OF-CEAS adopt selective optical feedback from cavity resonance to deliver efficient cavity injection. In this approach, a fraction of resonant light returns to laser when the laser emission is injected to the cavity and cavity resonance occurs. When the feedback field is in phase with laser radiation and has an appropriate feedback rate, it can be used to efficiently narrow the linewidth of the laser diode and also lock the laser frequency to cavity’s resonance frequency automatically [24]. Acting as an injection seeding, the feedback field will force laser emission narrower to lase on the precise frequency of the resonant cavity mode, yielding a narrower laser linewidth and an enhancement of laser injection. Therefore, the intensity of cavity transmission will be orders of magnitude bigger in comparison to traditional CEAS [25]. The laser radiation can maintain the frequency locking for a much longer time compared with the cavity ring-down time; hence the laser emission locking to the ordinal excited cavity modes can be achieved as the laser’s driving current is scanned linearly.

In general, OF-CEAS permits the peak value of TEM_00_ cavity mode to be measured, and obtain a better S/N ratio compared with conventional CEAS system. The sensitivity is improved by orders of magnitude. The OF-CEAS absorption spectrum is obtained within a limited spectral range by scanning the laser emission over hundreds of sequential cavity modes continuously, the recorded mode intensity (*I*) and the corresponding mode intensity without absorption (*I*_0_) are used to calculate the absorption coefficient *α*. Assuming that absorption of sample gas in a single pass is small compared to mirror losses [26], the OF-CEAS formula is:(3)$$\alpha =\frac{1-R}{L}(\sqrt{\frac{I_{0}}{I}}-1),$$
where the effective reflectivity (*R*) is determined by the cavity decay of “empty cavity”.

For some typical CEAS systems, the spectral resolution is usually limited by the FSR of the optical cavity when the laser frequency is jumping from one resonance to the next. In our experimental system, one of the end-mirrors (M2) has a specially designed structure, the piezoelectric ceramic transducer (PZT) mounted on M2 (shown as in Figure 1) can precisely modulate the cavity length to achieve cavity resonance, and the spectral resolution will not be limited by physical cavity length anymore. Thus, the frequency tuning resolution of the laser source will be the dominating factor, and the spectral resolution can be efficiently improved. In this approach, optical loss measurements without absorption are performed in advance by CRD technique per laser scan to calibrate the effective reflectivity (*R*).

### 2.2. Optical Setup

The established near-infrared OF-CEAS experimental setup is schematized in Figure 1. Here, the light source is a fibered DFB laser diode in a 14-pin butterfly package (SWLD-1531.5-15-PM, Allwave Lasers, Xi’an, China). It is housed in a universal laser diode mount (LM14S2, Thorlabs, Newton, NJ, USA), and its emission is centered at 1531.6 nm. It should be noted that this fibered diode does not include an optical isolator which would prevent the optical feedback from the V-shaped cavity.

A current deriver (LD202C, Thorlabs, Newton, NJ, USA) and a temperature controller (TED200C, Thorlabs, Newton, NJ, USA) are used to regulate the injection current and temperature of this laser diode, respectively. During a single laser scan, the operation temperature is constant, and a voltage ramp generated by a Dynamic Signal Acquisition and Generation device (NI-PCI-4461, 204.8 kS/s, National Instruments, Austin, TX, USA) is fed to the current deriver to achieve an accurate laser frequency scanning. The pigtail output from DFB laser diode is coupled into a single-mode polarization-maintaining (PM) fiber, followed by an optical switch which allows instantaneous switching off of the laser (10 ns). Then, the output of the optical switch is connected with the port1 of a PM fiber coupler (50:50, 1550 nm center wavelength), the port2 and port3 of this fiber coupler are connected to a fiber collimator and a photo-detector (PDA400, Thorlabs, Newton, NJ, USA), respectively. The collimated light from the fiber collimator is then injected into the V-shaped cavity at the incident mirror (M0). It should be noted that several PM fiber couplers with splitting ratio between 50:50 and 99:1 are prepared to adjust the OF ratio owing to the easy replacement.

In this experimental setup, the optical resonator is a typical V-shaped structure with two equal arms (250 mm) forming a folded angle of 7.4°. Glass-ceramic is applied as the cavity body for its ultralow expansion coefficient (~10^−8^/°C), and the cavity mirrors are fabricated with fused quartz (whose diameter is 25 mm). Among them, M0 is a plane mirror, M1 and M2 are spherical mirrors with 2 m radius, and M2 is mounted on a PZT to modulate the cavity length in a small region. The ultra-low loss coating (*R* ≈ 99.99% at 1530 ± 30 nm) lies in the central area (diameter is 10 mm) of the cavity mirrors. Two right-angle prisms with a sharp angle of 27° are optically glued on the backsides of M0 and M1, so that the etalon effect is eliminated. The cavity’s main body and three mirrors constitute an optical cavity length (L1 + L2) of 500 mm, and this V-shaped cavity can obtain the resonant optical feedback while avoiding the directly reflected beam of M0.

For our experimental setup, the maximum available spectral resolution is determined by FSR of the optical cavity if the cavity length is static, corresponding to a spectral resolution of 0.01 cm^−1^ (300 MHz) for this 50 cm long cavity. We enhance the spectral resolution by using simultaneous laser frequency tuning and cavity dither. Cavity mirror M2 has a novel geometry structure, and it is mounted on PZT1 to achieve precise cavity length modulation. In addition, the accurate wavenumber corresponding to the stepping injection current is calibrated in advance with a high precision wavemeter (WA-1500, EXFO, Quebec City, QC, Canada). In order to record the value of peak transmission of successive TEM_00_ cavity modes during a laser scanning, the optical path DFB-M1 should be arranged with a multiple of the cavity length (L1 + L2) based on the theoretical analysis [18], which is two times of cavity length in our case. When the injection current of DFB is scanned step to step, L2 is modulated by PZT1 to achieve cavity resonance and the laser-cavity distance is adjusted by PZT2 to obtain the appropriate OF phase for all modes dynamically. Another important control parameter for OF locking is the OF ratio. In this setup, the appropriate OF ratio is controlled by several PM fiber couplers with splitting ratios between 50:50 and 99:1. Weak feedback is applied to make sure that the efficient frequency locking range is much smaller than FSR of the optical cavity in this OF-CEAS.

The optical intensity signal transmitted by M1 is detected with a photodiode (PD1) with a high S/N ratio, and another photodiode (PD2) recording the incident laser intensity is used to normalize the absorption spectra. All these components mentioned above are placed on a vibration isolated optical bench. For automatic operation, an analogic feedback circuit for PZT2 regulation is utilized, which apply the characteristics of the cavity mode profile as presented in transmitted optical intensity signal during a laser scan to generate a real-time error signal [27]. The 24-bit Dynamic Signal Acquisition and Generation device (NI-PCI-4461, 204.8 kS/s) records the laser power and transmission signals during each laser scan.

The sample preparation for this setup is quite simple. All samples are exhaled into a 10 L Tedlar bag through a disposable mouthpiece and a Midisart-2000 air filter (Sartorius, Gottingen, Germany), which removes the air particulates and offers the ability of online sterilization. An ultra-high purity in-line gas filter is applied between the Tedlar bag and the cavity to prevent the organic macromolecules, which gives 3-nanometer particle filtering capability and high flow efficiency with minimum pressure drop. This V-cavity is fabricated to offer a small sample cell, and the internal volume in the cell is about 35 cm^3^. Measurements are performed with the sample gas flowing through the internal space of the V-shaped cavity continuously, where a pressure gauge is used to measure the pressure, and the flux is regulated to constant 0.1 L/min. A dry pump is used to control the pressure of internal space owing to the need of low sample gas pressure, for it is necessary to enhance the separation of absorption lines especially when multiple interference gases appear in exhaled human breath gas.

## 3. Results and Discussion

Measurement of the ring-down time (τ_0_) of the empty cavity is necessary before computing the absorption coefficient spectra of the sample gas. In this experimental setup, an efficient exponential decay cannot be obtained by shutting down the laser current driver due to its limited bandwidth (250 kHz). This procedure is accomplished by using a square pulse to make the driver of optical switch interrupt the laser instantaneously (10 ns). The laser is instantaneously switched off and the resonance light decays exponentially. The fast ring-down decay is recorded by a high sampling rate digitizer (CS320A, 100 MHz, Cleverscope, Auckland, New Zealand), and ring-down time was calculated by an exponential fit, as shown in Figure 2.

At 1531.6 nm the ring-down time was measured as *τ*_0_ = 7.75 µs with dry N_2_ flowing through the V-cavity, corresponding to a cavity finesse of 14,610 (effective absorption path length of 2.325 km). When the absorption loss of single-pass is small enough (*αL* << 1), the equivalent reflectivity (*R*) can be determined with the empty cavity ring-down time (*τ*_0_):(4)$$\frac{1-R}{L}=\frac{1}{c_{0}\tau_{0}},$$
here, c_0_ represents the speed of light in vacuum.

### 3.1. Measurement of 10 ppm of NH_3_ in N_2_

Ammonia has several absorption lines around 1531.6 nm. Before the measurement of exhaled human breath gas, the absorption spectrum of 10 ppm of NH_3_ in N_2_ at a total pressure of 0.1 atm and a temperature of 296 K is performed. This spectrum spans from 6528.91 cm^−1^ to 6530.66 cm^−1^, corresponding to a laser-current regulation from 40 to 120 mA at a working temperature of 27 ℃. During the measurement, the voltage signal generation, synchronous light intensity signal acquisition and transmission peak seeking for all frequency points in a single scan are controlled by a self-compiled Labview program. The cavity transmission values are transformed to absorption coefficient simultaneously using Equation 3 with (1-*R*) value determined from the ring-down time.

Figure 3a,b show the cross-section of NH_3_, H_2_O and CO_2_ at the temperature of 296 K and a total pressure of 1.0 atm and 0.1 atm, respectively. Absorption cross-section *σ* is simulated from the parameters given in the HITRAN database taking into account pressure and temperature [28]. As Figure 3a,b demonstrate, the absorption lines of NH_3_ are not easily identified at a total pressure of 1.0 atm, owing to the overlap of the neighboring absorption lines of ammonia itself, carbon dioxide and water vapor. By reducing the total pressure of sample gas, the linewidth of each absorption line turns to be narrower, and the interference caused by adjacent absorption lines is approximately eliminated. As shown in Figure 3b, the absorption line centered at 6528.77 cm^−1^ (cross-section *σ*_peak_ = 5.3944 × 10^−20^ cm^2^) is the optimal option for NH_3_ detection in this region. The absorption coefficient spectra of 10 ppm of NH_3_ in N_2_ computed from HITRAN 2012 data [29] are also shown for comparison in Figure 3c.

Figure 3c shows the comparison of measured absorption coefficient spectra of 10 ppm NH_3_ in N_2_ at a total pressure of 0.1 atm in a single scan against the simulations from HITRAN 2012 database. A good overall agreement is obtained in this spectral region, and the difference between HITRAN simulation and the measured spectrum is shown in the bottom residuals in Figure 3c, we attribute this mismatch to the accuracy of temperature and pressure control of sample gas, as well as the quality of HITRAN data. In Figure 3d, the fitting of this spectrum by Voigt profile and the fit residuals are presented, a minimum detectable absorption coefficient of *α*_min_ = 1.6 × 10^−9^ cm^−1^ is given by the standard deviation of residuals for the measured cavity ring-down time of *τ*_0_ = 7.75 µs. Aimed at the absorption line centered at 6528.77 cm^−1^ which has a cross-section *σ*_peak_ = 5.3944 × 10^−20^ cm^2^ at 296 K, using the relation *α*_min_ = *n*_min_*σ*_peak_, this *α*_min_ yields a detection limit of 12 ppb for NH_3_ at a total pressure of 0.1 atm and the temperature of 296 K. Noted that this *α*_min_ corresponds to a better detection limit of 6.8 ppb for NH_3_ at a total pressure of 1.0 atm, however, low sample pressure is necessary to eliminate the interference of other components for exhaled human breath gas analysis.

### 3.2. Measurement of Breath Gas

Finally, we demonstrate two spectra of exhaled human breath gas samples in Figure 4, which are taken from a healthy subject and a subject suffering from *Helicobacter pylori* infection. The samples of each subject were exhaled into a 10 L Tedlar bag through a disposable mouthpiece, and then the sample was regulated to flow through the V-shaped cavity at a flow rate of 0.1 L/min. The internal pressure is fixed to 0.1 atm by two pressure regulators, which are placed at the input and output of the cavity, respectively. The absorption spectra shown in Figure 4 indicate that the NH_3_ absorption line at 6528.77 cm^−1^ is sufficiently isolated from H_2_O and CO_2_ lines, which are the major interferences in exhaled human breath gas (about 4% CO_2_ and 3% H_2_O). The Voigt profile is applied to the fitting of each experimental absorption line, during the fitting process, all parameters of the Voigt profile are floated except the Doppler width, which is a fixed value for the gas with a specific temperature.

In the measurements of ammonia in exhaled human breath, eight subjects are invited, including six healthy adults and two patients suffering from *Helicobacter pylori* infection. For every subject, samples are acquired repeatedly to be measured. Results show that the concentration of ammonia in breath gas of healthy adult is in the range of 300–600 ppb, and 2.5–5 ppm for patients suffering from *Helicobacter pylori*. It should be noted that the concentration of NH_3_ is relevant to gender, age, lung capacity, diet habits and sampling time of day. Two representative subjects are illustrated in Figure 4 to demonstrate the capability of NH_3_ measurement in exhaled human breath. In Figure 4, using the relation α = n × σ, the concentrations of ammonia are determined as 433 ppb and 3.40 ppm for subject-A and subject-B, respectively. Results show that the excess concentration of ammonia is always discovered in breath gas of patients suffering from *Helicobacter pylori* compared to healthy subjects. A good reproducibility of the measurements is given, and a stability of ± 1.2% for long-term measurements of NH_3_ in exhaled human breath is obtained. In addition, the concentrations of H_2_O and CO_2_ can also be calculated in this spectral region, and the concentrations of H_2_O and CO_2_ are 2.80%, 2.91% and 3.11%, 2.98% for subject-A and subject-B, respectively. The minimum detectable absorption coefficient of α_min_ = 2.3 × 10^−9^ cm^−1^ corresponds to a detection limit of 17 ppb for NH_3_ at a total pressure of 0.1 atm at the temperature of 296 K. The higher sensitivity of this OF-CEAS instrument is achieved by the consistency of multiple measurements, after fitting the repeated absorption coefficient spectra, a better minimum detectable absorption coefficient of *α*_min_ = 6.0 × 10^−10^ cm^−1^ is acquired for an average of 16 scans, corresponding to a minimum detectable concentration of 4.5 ppb for NH_3_ in exhaled human breath gas. In addition, the spectra presented in Figure 4 show that sufficient frequency points are obtained to define the line shape even when the linewidth is much narrower at low pressure, the acquirable spectral resolution of this OF-CEAS experimental setup can be up to 0.002 cm^−1^.

## 4. Conclusions

We have presented a novel near-infrared OF-CEAS scheme with a V-shaped cavity which enables high sensitivity and enhanced spectral resolution measurement for NH_3_ in exhaled human breath. A fibered DFB laser diode is used as light source, the single-mode PM fiber is applied between DFB and cavity for light transmission. We found that this OF scheme requires only a few optical components, and is much more compact, robust and simple with respect to active electronic frequency locking (for example, the Pound-Drever-Hall (PDH) technique), this structure is better adapted for in-situ measurement. An optical switch allows instantaneous switching off of the laser (10 ns); thus the normalization of transmission intensity signals is accomplished continuously using the optical cavity loss determined by CRDS before each laser scan.

In the measurements of exhaled human breath, the 1σ standard deviation of the residuals yields a minimum detectable absorption coefficient of *α*_min_ = 2.3 × 10^−9^ cm^−1^ for a single scan within 5 s, corresponding to a detection limit of 17 ppb for NH_3_ at a total pressure of 0.1 atm and the temperature of 296 K. This steady experimental setup allows the detection limit down to 4.5 ppb for an average of 16 scans. Reducing the sample pressure is applied to narrow the absorption lines, better separation of NH_3_ to H_2_O and CO_2_ is achieved. Compared with other gas sensors, this OF-CEAS device achieves direct and selective measurement of an NH_3_ absorption line with high sensitivity and better spectral resolution, and the interference of neighboring lines is approximately eliminated. With dense frequency points to define the line profile, the calculated concentration is much more accurate. Some comparative tests show that the excess concentration of ammonia is always discovered in breath gas of patients suffering from *Helicobacter pylori* infection compared to healthy subjects. This work demonstrates the ability of our OF-CEAS to real-time breath gas analysis.

In general, this sensor presents high sensitivity and accurate discrimination during the detection of gas molecules. Owing to the easy replacement of fibered DFB laser diode, this setup can also be used for the measurement of other gas species by targeting the corresponding absorption lines. It should be noted that the line strength of the measured NH_3_ absorption line is one to two orders of magnitude weaker than the strongest line in the mid-infrared range. However, in the near-infrared spectral region, the DFB diode lasers and photodetectors are commercially available owing to the development of telecommunications. Besides, these devices also perform higher capability at room temperature compared with the devices working in the middle-infrared region. Therefore, this near-infrared setup is suitable for high sensitivity and relatively affordable sensors for clinical applications.

## Figures and Tables

**Figure 1 sensors-19-03686-f001:**
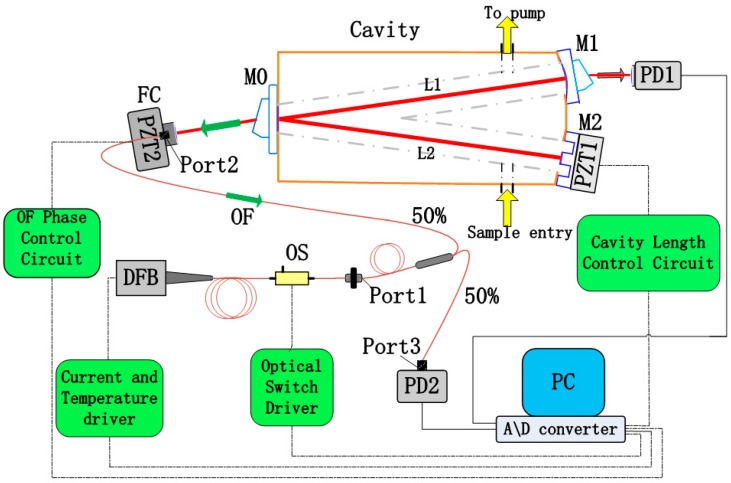
Scheme of the fibered OF-CEAS experimental setup with a fibered DFB laser diode. The laser-cavity path is constructed with fiber components, and the distance can be finely controlled by sub-wavelength displacements of the PZT2, which is connected to the PC to actively control the feedback phase. DFB: distributed feedback; M0-M2: mirrors; PD: photodetector; PZT: piezoelectric ceramic transducer; OF: optical feedback; FC: fiber collimator; OS: optical switch.

**Figure 2 sensors-19-03686-f002:**
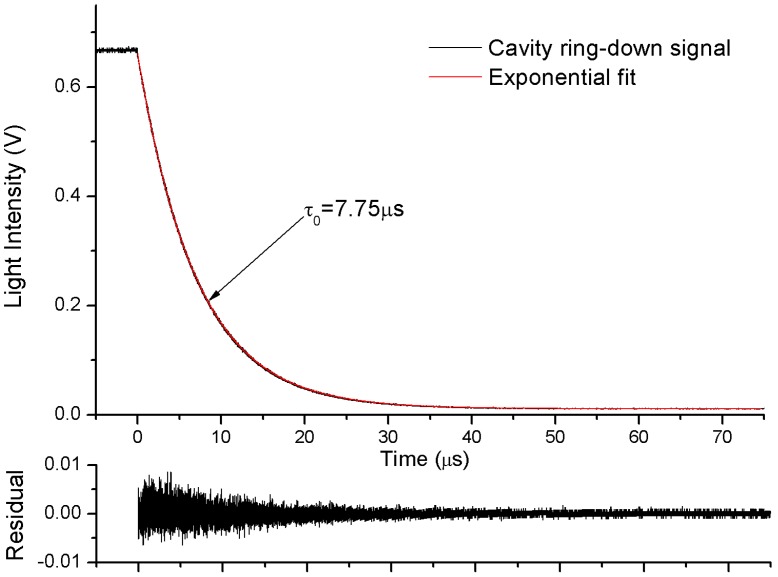
Typical cavity ring-down signals. The fitting residuals are shown in the lower panel.

**Figure 3 sensors-19-03686-f003:**
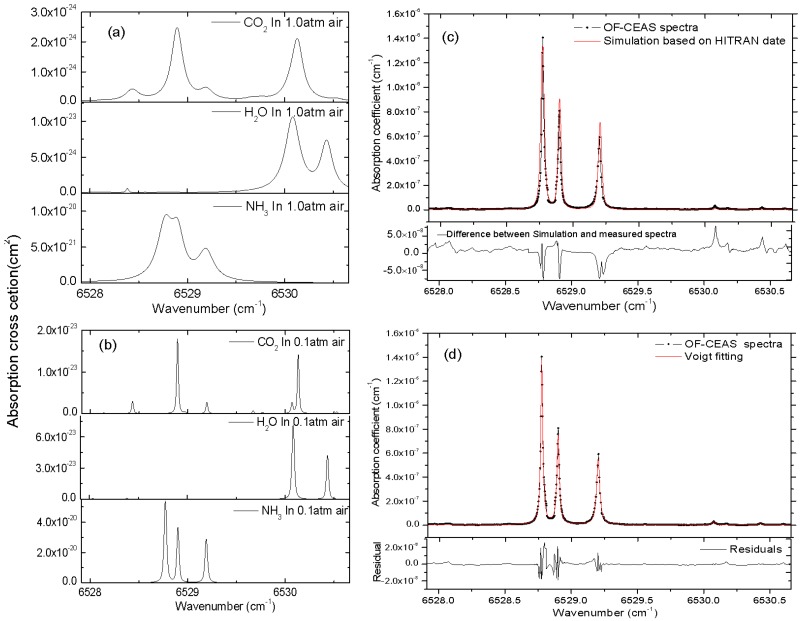
(**a**,**b**) represent the cross-section of NH_3_, H_2_O and CO_2_ at the temperature of 296 K and a total pressure of 1.0 atm and 0.1 atm, respectively. Measured absorption coefficient spectra of 10 ppm of NH_3_ in N_2_ at a total pressure of 0.1 atm (shown in black dot) together with absorption coefficient spectra computed from HITRAN 2012 data (shown as red line) are presented in (**c**). The fitting of the absorption spectra by the Voigt profile is also shown in (**d**), the standard deviation of the residuals gives a minimum absorption coefficient of *α*_min_ = 1.6×10^−9^ cm^−1^ in a single scan within 5 s.

**Figure 4 sensors-19-03686-f004:**
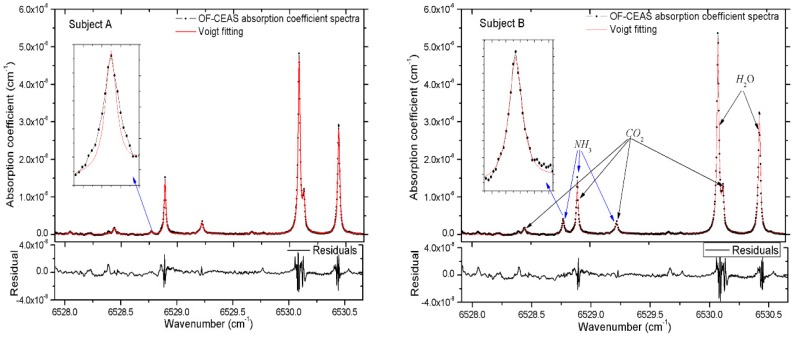
Absorption spectra of exhaled breath samples of two subjects. Subject A for a healthy adult aged 26, subject B for a 25 years old man suffering from *Helicobacter pylori* infection. The laser frequency is scanned over the range 6527.91–6530.66 cm^−1^. The absorption coefficient spectra is converted from the cavity transmission signals using Equation 3 in a single scan. The black dots represent a single laser scan achieved in 5 s, and the red line shows the Voigt fitting of all absorption lines including NH_3_, H_2_O and CO_2_. Below the spectra is the residuals to the fitting. The 1σ standard deviation of the residuals yields a minimum detectable absorption coefficient of *α*_min_ = 2.3 × 10^−9^ cm^−1^ for a single scan within 5 s.

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
