# Peer review of "Application of Near-Infrared Optical Feedback Cavity Enhanced Absorption Spectroscopy (OF-CEAS) to the Detection of Ammonia in Exhaled Human Breath"

_sensors, 2019, doi:10.3390/s19173686_

Round 1

Reviewer 1 Report

The article ‘Application of Near-infrared Optical Feedback Cavity Enhanced Absorption Spectroscopy (OF-CEAS) to the Detection of Ammonia in Exhaled Human Breath Gas' submitted by Zhifu Luo et al. describes an Optical feedback cavity enhanced spectrometer and its application in human breath detection. It has a certain novelty and its profile is completed. It will bring the attention of the community.  But I think, before accepted, a major revision is needed.

1.      There are numerous errors in English writing.

â… . Improper usage of preposition.

   For example, “for” in line 11 page 1 should be “of”. “to” in line 14 page 1 should be “for”.

â…¡. Improper usage of  punctuation

   For example, The first sentence in introduction. There should be a full stop between “monitoring”  and “it has been well known…” to separate the sentence.

â…¢. A lot of words you have used in your manuscript are inaccurate.

2.      In line 15 page 1, “eliminated” is inaccurate. You haven’t totally remove the interference from other molecular transition.

3.      In line 17 page 1, “frequency locked” is inaccurate.

4.      In line 20 page 1, “this stable system allows the spectra to be averaged over 10 laser scans to make the detection limit down to 4.5 ppb” should be “this stable system allows the detection limit  down to 4.5 ppb when the spectra to be averaged over 10 laser scans”

5.      In line 23 page 1, it is better to use cavity dither rather than “Cavity tuning”  

6.      In line 40 page 1, “particularly” should be “for example”

7.      Add more citation after line 52 page 2.

Most of your citation were published 10 years before. There are a bunch of groups to do human breath monitor by laser spectroscopy and optical feedback cavity enhanced spectroscopy.  

8.      “less intensity” in line 60 page 2 is misused.

9.      “weakening” in line 66 page 2 should be attenuation.

10.   In line 99, “which is extremely difficult for conventional CEAS and CRDS, therefore, this technique can effectively improve the spectral resolution and provides a better signal to-noise ratio (S/N)”. In CEAS and CRDS, it is a common method to scan the laser frequency and simultaneously dither the cavity to improve the spectra density. This is not your novelty. Moreover, CRDS is immune to intensity noise, so it can obtain an excellent SNR. I don’t think OF-CEAS can be better performed than CRDS. Can you explain it?

11.  Your introduction needs to be rewritten. The logic is so terrible. The content has a bad succession. It is better to re-arrange your sentence position and use more conjunction.

12.  It is better to re-conclude your novelty. To illustrate your novelty, it is better to compare your setup and method with the prior work.

13.   In line 113, “the novelty of OF CEAS is that the optical cavity apply three mirrors to constitute a “V-shaped” configuration.” The usage of V-shape cavity is not the essense of OF-CEAS. Moreover, researchers also have tried other type of cavity in OF CEAS.

14.  In figure 2, why the residual shows different noise behavior. In the beginning, it looks like a white noise, but after 25 us, it seems the bad DAQ resolution is the main noise source.

15.  In paragraph after 3.1, you have no sentence to describe the content in figure 3(a) and (b).

16.  In figure 4, can you analyse the reason of the different concentration results between the two samples?

Reviewer 2 Report

The authors describe an application of a optical feedback cavity enhanced absorption spectroscopy (OF-CEAS) for ammonia detection in exhaled breath. With appropriate optical feedback, a distributed feedback (DFB) diode laser is frequency locked to a V-shaped cavity with a free spectral range (FSR) of 300 MHz and a finesse of 14610. A minimum detectable absorption coefficient of αmin=2.3×10-9 cm^-1 is achieved in a single scan within 5 s, yielding a detection limit of 17ppb for NH3 at low pressure of 0.1 atm. Benefiting from the simultaneous laser frequency and cavity tuning, the absorption line profile is accurate for the high spectral resolution of 0.002 cm^-1. This work is promising and interesting, and the manuscript is well organized and written. The reviewer believes it is suited for publish in the Sensors after minor revision. 

Is there any pre-treatments for the exhaled breath samples? The authors should give details for the sample preparations.

Line 69, non-capitalized the first letter, "Where"

Line 70, ",."

Line 72, non-capitalized the first letter, "Here"

Line 132, non-capitalized the first letter, "Here"

Line 222, non-capitalized the first letter, "Here"

Reviewer 3 Report

Comments and suggestions are attached by the reviewer in: review_562496.pdf

Reviewer 4 Report

Manuscript Ref: sensors-562496

Review Report

Article title: Application of Near-infrared Optical Feedback Cavity Enhanced Absorption Spectroscopy (OF-CEAS) to the Detection of Ammonia in Exhaled Human Breath Gas

This manuscript describes an interesting work of the application of a high spectral resolution optical feedback cavity enhanced absorption spectroscopy to ammonia detection in the exhaled human breath gas with the aim of optimizing the main interference gases (CO2 and H2O) that are eliminated at the same time. This is an original and applicative idea. The manuscript is well presented.

Authors are made this manuscript selected for good work and I find a lot of interesting information included here.

However, there are some key information are missing, and some scientific explanation didn't provided. Therefore, it is not suitable to be published in its present form. Following revisions are suggested:

·        Line 40-41 ’’…exhaled breath gas of patients suffering from lung cancer’’ please introduce reference [Breathing Disorders Using Photoacoustics Gas Analyzer, Journal of Medical Imaging and Health Informatics, 6(8): 1893–1895, 2016,  https://doi.org/10.1166/jmihi.2016.1944] 

·        Line 51-52 ’’...methods based on laser absorption spectroscopy show a vibrant developing and applying for trace gas detection’’ please introduce the references [’Ultrasensitive, real-time trace gas detection using a high-power, multimode diode laser and cavity ringdown spectroscopy, Applied Optics, 55 (16): 4497-4504, (2016), https://doi.org/10.1364/AO.55.004497; Sub-parts-per-trillion level sensitivity in trace gas detection by cantilever-enhanced photo-acoustic spectroscopy, Scientific Reports volume 8:1848 (2018) and Ethylene Measurements from Sweet Fruits Flowers Using Photoacoustic Spectroscopy, Molecules, 2019 Mar; 24(6): 1144. doi: 10.3390/molecules24061144]

·        Line 76 ‘’ …renal failure and oral cavity disease’’, please complete the sentence with breath gas of patients suffering from mental disorders and introduce the references: [Laser scanning laser diode photoacoustic microscopy system, Photoacoustics, 2018, 9: 1–9. doi: 10.1016/j.pacs.2017.10.001; "Detection of ethylene by infrared spectroscopy in mental disorders", Rom Rep Phys Vol. 67(4): 1565-1569, 2015 and Fruit Quality Evaluation Using Spectroscopy Technology: A Review, Sensors (Basel). 2015 May; 15(5): 11889–11927. doi: 10.3390/s150511889]

·        The sampling procedure is unclear.

·        Lack of LODs, LOQs, reproducibility of the measurements.

·        Nothing about calibration of apparatus?

·        In general there are some grammatical mistakes, which should be carefully corrected. I suggest the author to spend some time to polishing the manuscript and to carefully correct some grammatical mistakes. At section 2: Methods, line 261 section 3.2 , Measurement of….. instead of  ‘’measurement of breath gas’’

Round 2

Reviewer 1 Report

I think the authors have replied my comments in detail. I recommend the manuscript to be accepted.

Author Response

Dear Reviewer: Thank you again for your comments concerning our manuscript entitled “Application of Near-infrared Optical Feedback Cavity Enhanced Absorption Spectroscopy (OF-CEAS) to the Detection of Ammonia in Exhaled Human Breath” (ID: sensors-562496). We tried our best to improve the manuscript and made some changes in the manuscript. In addition, we have our manuscript checked by a colleague who received her bachelor’s degree and doctorate in the United Kingdom. The grammar, spelling and punctuation in this manuscript are checked and modified carefully, all the changes are marked in red in revised manuscript. We appreciate for Reviewer’s warm work earnestly, and hope that the correction will meet with approval.

Reviewer 3 Report

The content corrections after initial review are satisfactory; significant English review is required before publication. Please check grammatical correctness.

Author Response

(The authors gave the same response as above.)
